# A national audit of facilities, human and material resources for the comprehensive management of diabetes in Ghana-A 2023 update

**Ernest Yorke**[1]*, **Josephine Akpalu**[1], **Gwendolyn de-Graft Johnson**[1], **Yacoba Atiase**[1], **Margaret Reynolds**[1], **Ruth Laryea**[1], **John Tetteh**[2], **Alfred E. Yawson**[2], **Albert G. B. Amoah**[1]

**1** Department of Medicine & Therapeutics, University of Ghana Medical School, College of Health Sciences, Accra, Ghana, **2** Department of Community Health, University of Ghana Medical School, College of Health Sciences, Accra, Ghana

* eyorke@ug.edu.gh

**Data Availability Statement:** All relevant data are within the manuscript and its Supporting Information files.

## Abstract

### Introduction

The human and material resources as well as the systems for managing diabetes in Africa are inadequate. This study or needs assessment, aimed at updating the human and material resources, identifying the gaps and unmet needs for comprehensive diabetes care in Ghana.

### Methods

We conducted a national audit of 122 facilities in all 16 administrative regions of Ghana. Information obtained covered areas on personnel and multidisciplinary teams, access to medications, access to laboratory services, financing, screening services, management of diabetes complications, and availability/use of diabetes guidelines or protocols. Data was analysed using STATA version 16.1. *P-values* <0.05 were set as significant.

### Results

Only 85(69.7%) out of the 122 surveyed facilities had a dedicated centre or service for diabetes care. Twenty-eight (23%) had trained diabetes doctors/specialists; and whilst most centres had ophthalmic nurses and dieticians, majority of them did not have trained diabetes educators (nurses), psychologists, ophthalmologists, podiatrists, and foot/vascular surgeons. Also, 13.9% had monofilaments, none could perform urine dipstick for microalbumin; 5 (4.1%) and just over 50% could perform laboratory microalbumin estimation and glycated haemoglobin, respectively. Access to and supply of human insulins was better than analogue insulin in most centres. Nearly 100% of the institutions surveyed had access to metformin and sulphonylurea with good to excellent supply in most cases, whilst access to Sodium Glucose Transporter-2 inhibitors and Glucagon-like peptide-1 analogues were low, and moderate for Dipeptidyl peptidase-4 inhibitors and thiazolidinediones. Majority of the health

**Funding:** This study funded jointly by the authors and a mid-career grant from the University of Ghana (UG-ORID 13th Call: MCG-002/12-20) to EY, JA, YA and AEY. The funder did not play any role in the study design, data collection and analysis, decision to publish, or preparation of the manuscript. Sponsor URL is https://orid.ug.edu.gh/

**Competing interests:** The authors have declared that no competing interests exist

facilities (95.1%) offered NHIS as payment mechanism for clients, whilst 68.0% and 30.3% of the patients paid for services using out-of-pocket and private insurance respectively. Fifteen facilities (12.3%) had Diabetes Support Groups in their locality and catchment areas.

## Conclusion

An urgent multisectoral collaboration, including prioritisation of resources at the facility level, to promote and achieve acceptable comprehensive diabetes care is required.

## Introduction

Global prevalence of diabetes among adults aged 20–79 years in 2021 was estimated by the International Diabetes Federation (IDF) to be 537 million, which is predicted to rise to 783 million by 2045, with the largest increases expected in sub-Saharan African countries [1]. Estimated figures by IDF for Ghana were 329,200 in 2021, representing about 2% of the adult population [1]. However, community-based studies suggest a much higher prevalence (6.46%; 95% CI: 4.66–8.26%) than stated [2].

The rising incidence and prevalence of especially type 2 diabetes are due to sedentary lifestyles, obesity, and poor dietary habits [3]. This would bring with it both microvascular (neuropathy, retinopathy and nephropathy) and macrovascular (stroke, myocardial infarction and peripheral vascular disease) complications [1, 3]. The expected increases in diabetes-related morbidity and mortality require corresponding improvements in human and material resources.

The doctor-patient ratio in Ghana has been improving steadily over the past few years with ratio in 2016 estimated to be about 1: 8,481 [4]. However, this is still below the recommended ratio of 1:1,000 by the World Health Organisation (WHO) [5]. Doctors with specialist knowledge or interest in diabetes in Ghana are conservatively put at 30, a situation that is woefully inadequate to achieve optimum care. The health system in Ghana is disproportionately skewed in favour of infectious diseases to the detriment of non-communicable diseases (NCDs) leading to a significant unmet need for diagnoses and treatment of NCDs in Ghana [6]. Also, community health participation and interventions for diabetes are low in Ghana [7].

In 1998, A study by Amoah *et al* revealed that only Korle-Bu Teaching Hospital (KBTH) had diabetologists among the 5 regional hospitals surveyed [8]. Dedicated centres for managing diabetes were also inadequate with only one functional diabetes clinic in KBTH. Currently, Ghana can boast of 5 public teaching hospitals, 10 regional hospitals, over 100 district and municipal hospitals, hundreds of health centres and over a 6000 small community service centres referred to as National Community Health Planning and Services (CHPS) compounds or zones [9]. Notwithstanding that CHPS compounds are not designated to treat diabetes, many of the other facilities do not have specialised centres for diabetes care. Most diabetes patients are therefore treated as part of the general clinic and hospital care. The study by Amoah *et al* [8] also revealed inadequacies in other trained staff, equipment and laboratory support services. Specifically, only 2 facilities had dieticians, 2 facilities had eye specialists, no facility had trained diabetes educators, there was no chiropodist or podiatrist in any of the facilities, clinical care equipment were lacking except for sphygmomanometers. Laboratory tests for managing diabetes were also found to be inadequate. Biochemical assays were largely available; however, glycated haemoglobin estimation could only be done at one centre, and no centre offered c-peptide, insulin autoantibodies and urine microalbumin estimation. Whilst the

supply of oral diabetes medications was relatively acceptable at the time, insulin supply was erratic. No facility had chronic haemodialysis service. At the time there was neither a diabetes advisory board nor guidelines for managing diabetes in Ghana at all levels in Ghana. Data on diabetes morbidity and mortality were scanty and there were 3 diabetes associations nationwide. The inadequate availability of healthcare personnel for managing diabetes and shortages of diabetes medications were confirmed in a later publication in 2022 [10].

Financing diabetes care is a challenge because of the high cost involved and this imposes a lot of economic burden and stress on patients with diabetes and chronic diseases in general [1, 10–12]. In 2015, a study of 40 households found out the extended family was the main resource of financial support for diabetes treatment and management [13]. The National Health Insurance Scheme (NHIS) established in 2006 has only partially addressed the financing gap in diabetes care [10, 13]. Whilst, metformin, sulphonylureas, thiazolidindiones and human insulins are covered under the NHIS, relatively newer medications like dipeptidyl peptidase IV (DPP IV) inhibitors, glucagon-like peptide1-1 (GLP-1) analogues, sodium glucose transporter 2 (SGLT 2) inhibitors and analogue insulins are not covered under the scheme [14]. The benefit package of NHIS tariffs also does not include cost of glucose monitoring meters and related devices, podiatry services, some aspects of eye care including laser therapy for diabetic retinopathy among others. Late reimbursement of costs to health facilities deny them of the much-needed resources, which also affects quality of care [14, 15]. Whilst the Standard Treatment Guidelines for Ghana has portions on the management of diabetes, it is not comprehensive and is also restricted for use at the primary and secondary levels. A truly national and comprehensive guidelines for the management of diabetes dedicated to all levels of care was unavailable at the time of this survey.

An assessment of the current state of diabetes care was urgently needed to identify the gaps and possible solutions for improvement. A system which is robust in terms of personnel, physical facilities, diabetes care equipment, adequate and affordable medicines and laboratory services, financing and governmental support is needed to address current and future challenges. This study or needs assessment sought to update the human and material resources as well as identify the gaps and unmet needs for comprehensive diabetes care in Ghana since the last update some 25 years ago.

## Materials and methods

### Study design and settings

This national audit was a cross-sectional study conducted from May 2021 to October 2021. It involved a total of 122 facilities in all 16 administrative regions of Ghana comprising 6 teaching (including 37 Military Hospital), 10 regional, 100 municipal/district hospitals, and 6 private/quasi-government hospitals (Fig 1).

### Recruitment, information gathering and ethical considerations

The Ghana Health Service (GHS) through the Non-communicable Disease Directorate provided an introductory letter granting the lead investigators and research assistants permission to undertake the survey in public health facilities, and to facilitate their entry and acceptance in these facilities. All public Teaching Hospitals, Regional Hospitals, most Municipal and District Hospitals in Ghana were contacted for the survey. The 37-Military Hospital Accra, a level-4 United Nations Hospital (and which also operates essentially as a Teaching Hospital) in Accra as well as some private and quasi-government health facilities were purposively selected (Fig 1). There were two teams for the nationwide assessment, one covering the southern half of the country, and the other the northern half of the country. They were trained over 3-day

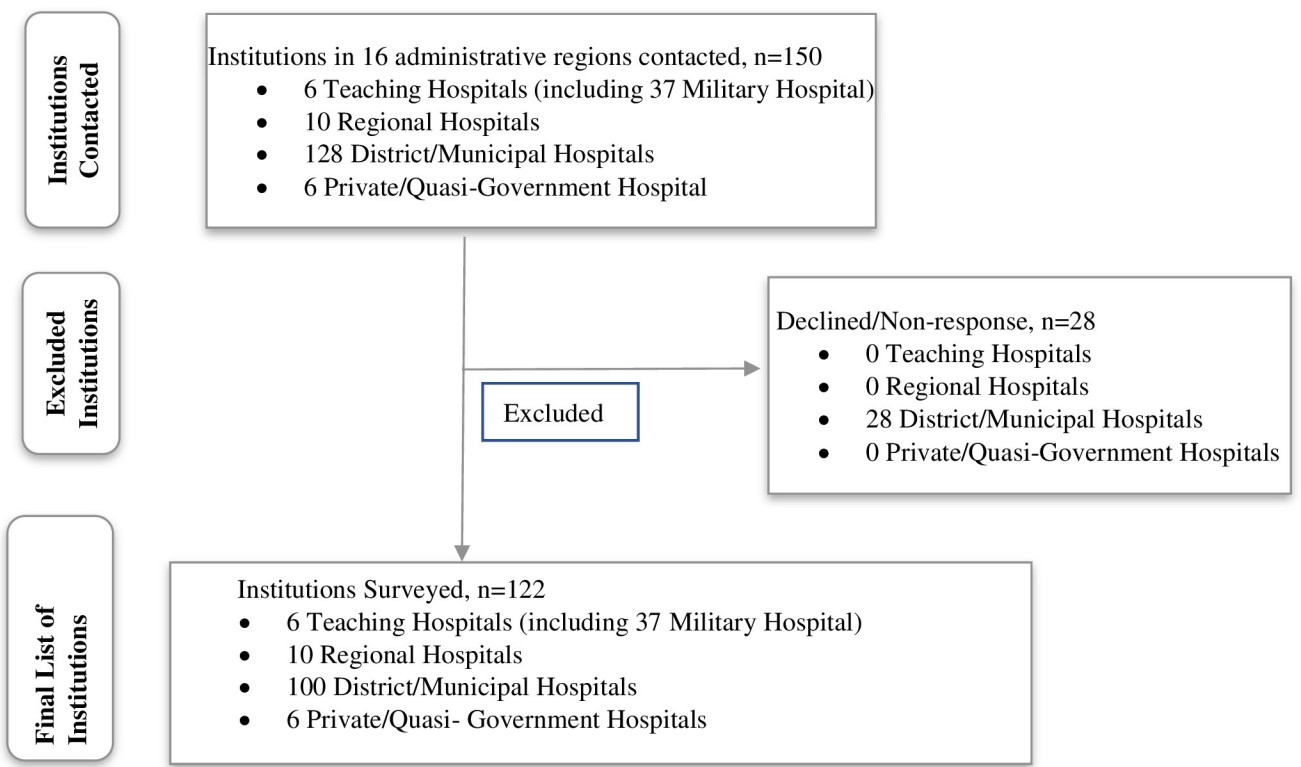

**Fig 1. Recruitment algorithm.**

period in community/facility entry protocols, selection of responsible respondents, questionnaire administration, data capture and entry.

Respondents were consented hospital administrators, medical directors and superintendents, and healthcare professionals involved in diabetes care in these facilities. A pretested questionnaire was used for data capturing and either administered in-person or by telephone.

*Study variables*: Information obtained centered on their current state of diabetes care in terms of facilities and resources as well as existing gaps in managing diabetes. Specific areas assessed included personnel, multidisciplinary teams (trained diabetes doctors/specialists, diabetes educators and nurses, ophthalmic nurses, psychologists, ophthalmologists, foot care specialists/podiatrists dieticians and foot or vascular surgeons) [16], access to medications, access to laboratory services, financing, screening services, management of diabetes complications, and availability/use of diabetes guidelines or protocols as well as support groups. Details of the specific items and questions are set out in the questionnaire attached as a supporting file (S1 File).

All participants provided written informed consent. Ethical and Protocol approval for the study and all the sites involved was sought from the University of Ghana College of Health Sciences Ethical and Protocol Review Committee with reference number CHS-Et/M2 5.15/2019-2020. It complied with the Helsinki Declaration of 1964 (Revised 2013) on human experimentation.

## Data handling and statistical methods

Data handling codes were assigned to respondents for confidentiality. Only the principal investigator matched codes to respondents. Questionnaires were stored in a safety cabinet under lock and key whilst collected data were entered and stored into an electronic database

which was password protected. Data were entered into Microsoft Excel version 12 and imported into the STATA version 16.1 for analysis. Data were analysed according to the thematic arrears outlined in the data extraction tools. The availability of specified services, medications and personnel were quantitatively tallied, while the supply of the medications were the subjective views of the respondents described qualitatively as 'below average', average', 'good' or excellent'. Descriptive statistics of the responses were summarized in appropriate frequency tables and flow charts. Categorical variables were summarized as frequencies and percentages whilst continuous variables were summarized as means. Responses were examined for internal consistencies. Significance level was set at a p-value less than 0.05.

## Results

In all, one-hundred and twenty-two out of a total of 150 health facilities contacted responded, giving a response rate of 81.3% (Fig 1). This comprised of 6 teaching (4.9%), 10 regional (8.2%), 100 district/municipal (82%) and 6 private/quasi-government (4.9%) hospitals and health facilities. All the 16 administrative regions of Ghana were represented with Greater Accra, Eastern and Ashanti regions contributing 16.2% (n = 19), 15.4% (n = 18) and 12.8% (n = 15) respectively (Table 1).

Only 85 (69.7%) out of the 122 surveyed facilities had a dedicated centre or service for diabetes care (Table 2), with 6 (7.0%), 10 (11.8%) and 4 (4.7%) being from the teaching, regional and private/quasi-government health facilities respectively. Twenty-eight (23%) had trained diabetes doctors/specialists with a total of 9 (32.1%) from the teaching and regional hospitals. Whilst most centres had ophthalmic nurses (87.7%) and dieticians (64%), majority of them did not have trained diabetes educators (nurses) (69.7%), psychologists (95%),

**Table 1. Type of health facilities and geographical locations.**

| Type of health facility | Frequency (n = 122) | % |
|---|---|---|
| Teaching Hospitals | 6 | 5.7 |
| Regional Hospitals | 10 | 4.1 |
| District/Municipal Hospitals | 100 | 86.1 |
| Private/Quasi-Governments | 6 | 4.1 |
| **Region** | | |
| Ahafo | 9 | 7.7 |
| Ashanti | 15 | 12.8 |
| Bono East | 3 | 2.6 |
| Bono | 7 | 6.0 |
| Central | 9 | 7.7 |
| Eastern | 18 | 15.4 |
| Greater Accra | 19 | 16.2 |
| North-East | 2 | 1.7 |
| Northern | 3 | 2.6 |
| Oti | 1 | 0.9 |
| Savannah | 2 | 1.7 |
| Upper East | 10 | 8.5 |
| Upper West | 7 | 6.0 |
| Volta | 4 | 3.4 |
| Western North | 6 | 5.1 |
| Western | 2 | 1.7 |

**Table 2.** Availability of diabetes care centres and multi-disciplinary care team.

| Variable | Response | | Min-Max | Mean±SD | Median(IQR) |
|---|---|---|---|---|---|
| | No n(%) | Yes n(%) | | | |
| Availability of a Dedicated Diabetes Centre/ Care | 37(30.3) | 85(69.7) | NA | | |
| Trained Diabetes Doctors/Specialists | 94(77.0) | 28(23.0) | 1–8 | 2.78±2.17 | 5(1–4) |
| Diabetes educators | 85(69.7) | 37(30.3) | 1–4 | 1.9±1.11 | 1.5(1–3) |
| Ophthalmic nurses | 15(12.3) | 107(87.7) | 1–6 | 2.28±1.08 | 2(2–3) |
| Psychologists | 95(77.9) | 27(22.1) | 1–15 | 2.1±0.65 | 2(2–4) |
| Ophthalmologist | 91(74.6) | 31(25.4) | 1–17 | 2.2±0.65 | 2(2–4) |
| Podiatrists/ foot care specialists | 116(95.1) | 6(4.9) | 1–2 | 1.4±0.55 | 1(1–2) |
| Foot surgeon(s) / Vascular surgeon(s) | 113(92.6) | 9(7.4) | 1–4 | 1.7±1.09 | 1(1–2) |
| Dietician(s) | 58(47.5) | 64(52.5) | 1–20 | 2.1±0.67 | 3(2–4) |

**Availability of variables by facility type**

| Variable | Teaching Hospitals n(%) | Regional Hospitals n(%) | Private/Quasi-Governments n(%) | District/Municipal Hospitals n(%) |
|---|---|---|---|---|
| **Availability of a Dedicated Diabetes Centre/ Care** | 6(100.0) | 10(100.0) | 6(100.0) | 63(63.0) |
| **Trained Diabetes Doctors/Specialists** | 6(100.0) | 8(80.0) | 4(66.7) | 10(10.0) |
| **Diabetes educators** | 6(100.0) | 10(100.0) | 6(100.0) | 15(15.0) |
| **Ophthalmic Nurses** | 6(100.0) | 10(100.0) | 4(66.7) | 87(87.0) |
| **Psychologists** | 6(100.0) | 10(100.0) | 4(66.7) | 7(7.0) |
| **Ophthalmologist** | 6(100.0) | 10(100.0) | 2(33.3) | 13(13.0) |
| **Podiatrists/ foot care specialists** | 5(83.3) | 0(0.00) | 1(16.7) | 0(0.0) |
| **Foot surgeon(s) / Vascular surgeon(s)** | 6(100.0) | 2(20.0) | 1(16.7) | 0(0.0) |
| **Dietician(s)** | 6(100.0) | 10(100.0) | 6(100.0) | (42.0)42 |

Min: minimum; Max: maximum; SD: standard deviation; IQR: inter quartile range

ophthalmologists (74.6%), podiatrists/ foot care specialists (95.1%) and foot/vascular surgeons (92.6%). The teaching hospitals performed better in terms of these human resources, followed by the regional hospitals, quasi-government/private and then district/municipal hospital in that order (Table 2).

Whilst majority of facilities (about 90% or more) has sphygmomanometers, glucose monitoring devices, weighing scales and ophthalmoscopes, only 1.6% and 13.9% had biothesiometers and monofilaments respectively (Table 3). Of note, all the six teaching hospitals had monofilaments for assessing neuropathy, and the 2 institutions with biothesiometers were also teaching hospitals. None of the institutions could perform urine dipstick for microalbumin, and only 5 (4.1%) could perform laboratory microalbumin estimation. Just over 50% of the institutions surveyed could perform the very important glycated haemoglobin, however, all the teaching, regional and quasi-government/private hospitals could perform it; whilst only 5 (4.1%) could perform insulin autoantibodies and c-peptide estimation. Majority (>70%) were able to perform lipid and renal function tests (Table 3). All the teaching, regional and quasi-government/private hospitals could perform glycated haemoglobin.

The majority (>80%) of the health facilities had access to soluble and premix human insulins with a good to excellent supply in most instances. Only 36.9% had access to or use NPH insulin. Access to fast-acting analogue insulin, premixed analogue, long-acting and ultra-long-acting insulins were recorded in 24.6%, 30.3%, 35.2% and 15.6% out of the 122 institutions surveyed (Table 4). Nearly 100% of the institutions surveyed had access to metformin and

**Table 3. Equipment and laboratory services available for diabetes care.**

| Equipment | Absent n(%) | Present n(%) | |
|---|---|---|---|
| Sphygmomanometers | 1(0.8) | 121(99.2) | |
| Glucometers | 7(5.7) | 115(94.3) | |
| Ophthalmoscopes | 14(11.5) | 108(88.5) | |
| Monofilament | 105(86.1) | 17(13.9) | |
| Biothesiometer | 120(98.4) | 2(1.6) | |
| Weighing scales | 2(1.6) | 120(98.4) | |
| **Laboratory Services Available** | | | |
| Full blood count | 9(7.4) | 113(92.6) | |
| Urine dipstick for microalbumin | 122 (100) | 0(0.0) | |
| Urine dipstick for proteinuria | 8(6.6) | 114(93.4) | |
| Laboratory microalbumin estimation | 117 (95.9) | 5(4.1) | |
| Glycated Haemoglobin | 54(44.3) | 68(55.7) | |
| Lipids | 32(26.2) | 90(73.8) | |
| Renal function test | 34(27.9) | 88(72.1) | |
| Insulin autoantibodies | 117 (95.9) | 5(4.1) | |
| C-peptide estimation | 117 (95.9) | 5(4.1) | |
| Liver function Test | 35(28.7) | 87(71.3) | |

**Availability of selected variables by facility type**

| Variable | Monofilament n(%) | Biothesiometer n(%) | Glycated Haemoglobin n(%) |
|---|---|---|---|
| Teaching Hospital | 6(100.0) | 2(28.5) | 6(100.0) |
| Regional Hospital | 6(60.0) | 0(0.0) | 10(100) |
| Private/Quasi-Government | 4(66.7.0) | 0(0.0) | 6(100.0) |
| District/Municipal Hospital | 1(1.0) | 0(0.0) | 46(46.0) |

sulphonylurea with good to excellent supply in most cases, whilst access to SGLT-2 inhibitors and GLP-1 analogues were low, (17.2% and 9.8% respectively); and 23.0% and 38.0% of the health facilities had access to DPP-4 inhibitors and thiazolidinediones respectively (Table 4). Details of the availability of these medications by facility type has also been set out in Table 4.

Majority of the health facilities (95.1%) offered NHIS as payment mechanism for clients, whilst 68.0% and 30.3% of the patients paid for services using cash-and-carry and private insurance respectively, according to the facilities surveyed. One-hundred and twelve out of the 122 facilities (91.8%) surveyed used the Standard Treatment Guidelines (STGs) in their practice whilst 12.3% used a combination of either STGs, local, regional, or international guidelines (Table 5). Only 15 (12.3%) and 12 (9.8%) of the health facilities had Diabetes Support Groups and Associations respectively in their locality and catchment areas.

## Discussion

This study seeks to update the audit done 25 years go by Amoah *et al* [8], which characterised facilities and resources for diabetes care in regional hospitals in southern Ghana. It found inadequate dedicated centres for diabetes care, limited laboratory tests, equipment, and poor supply of medications especially insulin. In many ways, this study is more comprehensive: it covered all 10 existing regional hospitals, most district and municipal hospitals, all teaching hospitals including 37 military hospital, and some selected private and quasi-government hospitals.

**Table 4. Medication availability and supply to the health facilities.**

| | | Supply | | | |
|---|---|---|---|---|---|
| Medication Availability | Frequency n(%) | Below average n(%) | Average n(%) | Good n(%) | Excellent n(%) |
| **Soluble Insulin** | | | | | |
| No | 3(2.5) | | | | |
| Yes | 119(97.5) | 7 (5.9) | 21(17.7) | 65(54.6) | 26(21.8) |
| **NPH** | | | | | |
| No | 77(63.1) | | | | |
| Yes | 45(36.9) | 2(4.4) | 10(22.2) | 26(57.8) | 7(15.6) |
| **Premixed human Insulin** | | | | | |
| No | 19(15.6) | 3(2.9) | 20(19.4) | 64(62.1) | 16(15.5) |
| Yes | 103(84.4) | | | | |
| **Fast-acting analogue Insulin** | | | | | |
| No | 92(75.4) | | | | |
| Yes | 30(24.6) | 5(16.7) | 7(23.3) | 15(50.0) | 3(10.0) |
| **Premixed analogue insulin** | | | | | |
| No | 85(69.7) | | | | |
| Yes | 37(30.3) | 3(8.1) | 9(24.3) | 18(48.6) | 7(18.9) |
| **Long-acting Insulin** | | | | | |
| No | 79(64.8) | | | | |
| Yes | 43(35.2) | 6(14.0) | 8(18.6) | 19(44.2) | 10(23.3) |
| **Ultra Long-acting Insulin (Degludec/Tresiba)** | | | | | |
| No | 103(84.4) | | | | |
| Yes | 19(15.6) | 5(26.3) | 4(21.1) | 7(36.8) | 3(15.8) |
| **Metformin** | | | | | |
| No | 1(0.8) | | | | |
| Yes | 121(99.2) | 1(0.8) | 9(7.4) | 67(55.4) | 44(36.4) |
| **Sulphonylurea** | | | | | |
| No | 3(2.5) | | | | |
| Yes | 119(97.5) | 2(1.7) | 8(6.7) | 71(59.7) | 38(31.9) |
| **DPP-4 Inhibitors** | | | | | |
| No | 94(77.0) | | | | |
| Yes | 28(23.0) | 4(14.3) | 8(28.6) | 11(39.3) | 5(17.9) |
| **SGLT-2 Inhibitors** | | | | | |
| No | 101(82.8) | | | | |
| Yes | 21(17.2) | 4(19.0) | 3(14.3) | 10(47.6) | 4(19.0) |
| **Thiazolidinediones** | | | | | |
| No | 84(68.9) | | | | |
| Yes | 38(31.1) | 5(13.2) | 9(23.7) | 14(36.8) | 10(26.3) |
| **GLP-1 analogues** | | | | | |
| No | 110(90.2) | | | | |
| Yes | 12(9.8) | 5(41.7) | 3(25.0) | 4(33.3) | |

| Medication availability by facility type | | | | |
|---|---|---|---|---|
| Variables | Teaching Hospitals | Regional Hospitals | Private/Quasi-Government Hospitals | District/Municipal Hospital |
| Soluble insulin | 6(100.0) | 10(100.0) | 6(100.0) | 97(97.0) |
| NPH Insulin | 4(66.7) | 7(70.0) | 4(80.0) | 30(30.0) |
| Premix human insulin | 6(100.0) | 10(100.0) | 6(100.0) | 81(81.0) |

*(Continued)*

**Table 4.** (Continued)

| Supply | | | | |
|---|---|---|---|---|
| **Fast-acting analogue insulin** | 6(100.0) | 6(60.0) | 5(100.0) | 13(13.0) |
| **Premixed analogue insulin** | 6(100.0) | 6(60.0) | 6(100.0) | 19(19.0) |
| **Long-acting Insulin** | 6(100.0) | 5(50.0) | 5(83.3) | 27(27.0) |
| **Ultra long-acting Insulin** | 6(100.0) | 7(70.0) | 6(100.0) | 0(0.0) |
| **Metformin** | 6(100.0) | 10(100.0) | 6(100.0) | 99(99.0) |
| **Sulphonylurea** | 6(100.0) | 6(100.0) | 10(100.0) | 6(100.0) |
| **DPP-4 Inhibitors** | 6(100.0) | 8(80.0) | 5(83.3.0) | 9(9.0) |
| **SGLT-2 Inhibitors** | 6(100.0) | 6(60.0) | 6(100.0) | 3(3.0) |
| **Thiazolidinediones** | 6(100.0) | 10(100.0) | 5(83.3) | 17(17.0) |
| **GLP-1 analogues** | 5(83.3) | 0(0.0) | 6(100.0) | 1(1.0) |

NPH: Neutral Protamine Hagedorn; SGLT-2: Sodium-Glucose Cotranporter-2; DPP-4: Dipeptidyl Peptidase 4; GLP-1: Glucagon-like Peptide-1

Approximately two out of three of the 122 surveyed facilities had a dedicated centre or service for diabetes care. Whilst this is an improvement, a lot more effort needs to be made to establish such centres, especially in district and municipal hospitals to improve access to diabetes care. In the interim the Ghana Health Service Wellness Clinics concept [17] could be strengthened to provide diabetes care services to patients.

It was noteworthy that most institutions that were surveyed had dieticians and ophthalmic nurses, a situation that would enhance non-pharmacological control of diabetes and screening for eye complications respectively. The availability of trained diabetes nurses/educators, psychologists, ophthalmologists remain a challenge in most centres except the teaching and regional hospitals. Very few centres had podiatrists/ foot care specialists and foot/vascular surgeons. Foot care is an important aspect of diabetes care in preventing and managing diabetes-related foot problems. New strategies such as curriculum development, training and funding

**Table 5.** Payment schemes, guidelines and protocols and support groups.

| Variable | No<br>n(%) | Yes<br>n(%) |
|---|---|---|
| **Payment Scheme** | | |
| **NHIS** | 6(4.9) | 116(95.1) |
| **Private Insurance** | 85(69.7) | 37(30.3) |
| **Cash and Carry** | 39(32.0) | 83(68.0) |
| **Guidelines and Protocols** | | |
| **Local** | 107(87.7) | 15(12.3) |
| **Regional** | 119(97.5) | 3(2.5) |
| **Standard Treatment Guidelines** | 10(8.2) | 112(91.8) |
| **National, other** | 109(89.3) | 13(10.7) |
| **International** | 95(77.9) | 27(22.1) |
| **Combination** | 93(76.2) | 29(23.8) |
| **Diabetes Association & Diabetes Support Groups (at Facility or Catchment Area)** | | |
| **Diabetes Association** | 110(90.2) | 12(9.8) |
| **Diabetes Support Groups** | 107(87.7) | 15(12.3) |

NHIS: National Health Insurance Scheme

are urgently needed to produce and deploy more footcare specialists, as well as foot and vascular surgeons.

Whilst most centres had basic equipment like sphygmomanometers, glucose monitoring devices, weighing scales and ophthalmoscopes, only two teaching hospitals and one district hospital had biothesiometers and monofilaments respectively. The absence of the latter two makes the detection and subsequent management of diabetes neuropathy difficult. The widespread ability to perform basic biochemical tests such as renal and lipid function tests helps the detection and management of important co-morbid conditions of renal failure and dyslipidaemia. Efforts must however be made to procure the facilities to detect or estimate urine microalbumin, a very early sign of diabetic nephropathy. The situation where glycated haemoglobin, which is needed for diagnosis in some instances and crucially for monitoring, was available in just under 50% of district and municipal hospitals is unacceptable [18]. This situation must improve, and in the least, with the provision of affordable point-of-care devices for the measurement of HbA1c. Healthcare professionals must be educated on the need to periodically request HbA1c measurement to ascertain the level of glucose control in their patients. They must also engage the managers of such institutions to procure such equipment and devices.

Compared to previous findings, access to and supply of affordable human insulins, metformin and sulphonylurea remain good in most facilities across the country [8]. Efforts must be made to improve the supply of analogue insulins, SGLT-2 inhibitors and GLP-1 analogues, mostly for the district and municipal hospitals. Although, more expensive, the latter three offers some advantages in terms of less risk of hypoglycaemia and weight gain (analogue insulins), weight loss and cardiovascular benefits (SGLT-2 inhibitors and GLP-1 analogues) [19, 20]. Price reduction strategies such as waiver of import taxes and duties could help in making these relatively newer medications more accessible to patients.

A lot of the patients paid for services using cash-and-carry (out-of-pocket payments). Financing diabetes care is a challenge because of the high cost involved, and this imposes a lot of stress on sufferers of diabetes and chronic diseases in general [1, 11]. In 2015, a study of 40 households found out the extended family was the main resource of financial support for diabetes treatment and management [13]. Whilst access to NHIS as a payment mechanism has improved since its establishment in 2006, it has only partially addressed the financing gap in diabetes care [15]. Whilst human insulins and older oral antiglycaemic agents such as metformin and sulphonylurea are covered by NHIS, relatively newer ones like dipeptidyl peptidase-4 (DPP-4) inhibitors, glucagon-like peptide-1 (GLP-1) analogues, sodium glucose transporter 2 (SGLT 2) inhibitors and analogue insulins are not covered by under the scheme [14]. The benefit package of NHIS tariffs also does not include cost of glucose monitoring meters and related devices, many relevant laboratory tests (such as glycated haemoglobin and urine microalbumin estimation). Podiatry services, some aspects of eye care including laser therapy for diabetic retinopathy among others are also not covered by the NHIS. Other problems that have bedeviled the NHIS scheme are unrealistic tariffs for services as well as late reimbursement of costs to health facilities, which denies them of the much-needed resources, which also affects quality of care [14, 15].

Most of the facilities at the primary and secondary levels of care (health centres, district, and regional hospitals) surveyed used the Standard Treatment Guidelines (STG) as guide for managing diabetes. Since the audit in 1998 [8], Ghana still does not have a comprehensive national guidelines for diabetes care. The ongoing effort by the Ministry of Health, GHS and partners like the Diabetes Endocrine & Metabolic Society of Ghana (DEMSoG) to develop one must be expedited. In the absence of specialist or trained personnel in diabetes, guidelines help to decentralize diabetes best practices and across the country. It also becomes a tool for advocacy to improve diabetes care by stakeholders and healthcare practitioners.

Diabetes support group and associations provide patient support, counselling, advocacy and sometimes free or affordable consumables and medications such as glucose monitoring devices and insulin. Their formation and must be encouraged and supported to grow in many localities across the country.

## Study limitations

This study could not survey every healthcare facility in Ghana, especially, private health facilities. However, efforts were made to cover as widely as possible in terms of geographical distribution and categories of facility. Nonetheless, we believe the findings are generalizable for Ghana. There may have been some recall bias for those who responded by telephone. This possibility was reduced by allowing the respondents enough time to manually take stock of their available resources before scheduling an interview.

## Implications of study findings

The study has provided an up-to-date and truly national information on the human and material resources available for diabetes care in Ghana. It will serve an important document for stakeholders who may be interested in promoting comprehensive diabetes care in Ghana. It will also serve as an advocacy tool to improve funding, early detection, personnel, laboratory facilities, medication supply, detection, and management of diabetes-related complications.

## Conclusions

This study sought to update the audit done 25 years ago by Amoah *et al* [8], which characterised facilities and resources for diabetes care in regional hospitals in southern Ghana. Whilst a lot of improvement in human and material resources were noted, significant gaps remain, which requires urgent multisectoral collaboration to promote and achieve acceptable comprehensive diabetes care in Ghana. The non-communicable diseases policy of Ghana must be strengthened and funded to achieve its set goals in general, and in particular, the aspects that deals with diabetes care.

## Supporting information

**S1 File.**
(PDF)

**S2 File.**
(DOCX)

## Acknowledgments

We thank the Diabetes Endocrine and Metabolic Society of Ghana (DEMSoG) and the Ghana Health Service for their endorsement of the project.

## Author Contributions

**Conceptualization:** Ernest Yorke, Josephine Akpalu, Yacoba Atiase, Alfred E. Yawson, Albert G. B. Amoah.

**Data curation:** Ernest Yorke, Josephine Akpalu, Gwendolyn de-Graft Johnson, Margaret Reynolds, Ruth Laryea, John Tetteh.

**Formal analysis:** Ernest Yorke, Gwendolyn de-Graft Johnson, Yacoba Atiase, Margaret Reynolds, Ruth Laryea, John Tetteh, Albert G. B. Amoah.

**Funding acquisition:** Ernest Yorke, Josephine Akpalu, Yacoba Atiase, Alfred E. Yawson.

**Investigation:** Ernest Yorke, Josephine Akpalu, Gwendolyn de-Graft Johnson, Margaret Reynolds, John Tetteh.

**Methodology:** Ernest Yorke, Josephine Akpalu, Gwendolyn de-Graft Johnson, Yacoba Atiase, Margaret Reynolds, Ruth Laryea, John Tetteh, Alfred E. Yawson, Albert G. B. Amoah.

**Project administration:** Ernest Yorke, Gwendolyn de-Graft Johnson, Margaret Reynolds, Ruth Laryea, Albert G. B. Amoah.

**Resources:** Ernest Yorke, Josephine Akpalu, Yacoba Atiase, Alfred E. Yawson, Albert G. B. Amoah.

**Software:** John Tetteh.

**Supervision:** Ernest Yorke, Josephine Akpalu, Gwendolyn de-Graft Johnson, Yacoba Atiase, Margaret Reynolds, Ruth Laryea, Albert G. B. Amoah.

**Validation:** Ernest Yorke, Josephine Akpalu, Ruth Laryea, John Tetteh.

**Visualization:** Ernest Yorke, Alfred E. Yawson, Albert G. B. Amoah.

**Writing – original draft:** Ernest Yorke, Gwendolyn de-Graft Johnson.

**Writing – review & editing:** Ernest Yorke, Josephine Akpalu, Gwendolyn de-Graft Johnson, Yacoba Atiase, Margaret Reynolds, Ruth Laryea, John Tetteh, Alfred E. Yawson, Albert G. B. Amoah.

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
