## [Editor Report · Decision Letter 0]

24 Oct 2023

PONE-D-23-31002A national audit of facilities, human and material resources for the comprehensive management of diabetes in Ghana-A 2023 updatePLOS ONE

Dear Dr. Yorke,

Thank you for submitting your manuscript to PLOS ONE. After careful consideration, we feel that it has merit but does not fully meet PLOS ONE’s publication criteria as it currently stands. Therefore, we invite you to submit a revised version of the manuscript that addresses the points raised during the review process.

The authors should strengthen the background, methods and discussion of the findings. I have made extensive comments in the attached pdf. 

We look forward to receiving your revised manuscript.

Kind regards,

Sandra Boatemaa Kushitor, Ph.D.

Academic Editor

PLOS ONE

Journal Requirements:

Additional Editor Comments:

Dear Author,

Thank you very much for submitting this important work to PLOS One.

In my review of the submission, I noticed that the manuscript does not speak to the global burden of diabetes prevalence and care. The methods of the manuscript need to be strengthened. You must add a section on study variables and explain with references why the variables you assessed were selected.

Kindly revise the submission so that I will invite reviewers for the peer review process to begin.

---

## [Author Response · Author response to Decision Letter 0]

26 Nov 2023

Responses to Reviewers

1. Comment: The authors must add a section on the prevalence of T2D in Ghana. This suggestion has been effected. The new sentences now read as: 

Global prevalence of diabetes among adults aged 20-79 years in 2021 was estimated by the International Diabetes Federation (IDF) to be 537 million, which is predicted to rise to 783 million by 2045, with the largest increases expected in sub-Saharan African countries [1]. Estimated figures by IDF for Ghana were 329,200 in 2021, representing about 2% of the adult population [1]. However, community-based studies suggest a much higher prevalence than stated [2].

The rising incidence and prevalence of especially type 2 diabetes are due to sedentary lifestyles, obesity, and poor dietary habits [3]. This would bring with it both microvascular (neuropathy, retinopathy and nephropathy) and macrovascular (stroke, myocardial infarction and peripheral vascular disease) complications [1, 3]. The expected increases in diabetes-related morbidity and mortality require corresponding improvements in human and material resources. [page 3]

2. Comment:Provide reference for the statement “The incidence and prevalence of especially type 2 diabetes are increasing due to sedentary lifestyles, obesity, and poor dietary habits This suggestion has been effected. The new sentences now read as: 

The rising incidence and prevalence of especially type 2 diabetes are due to sedentary lifestyles, obesity, and poor dietary habits [3]. [page 3]

3. Comment:Kindly situate this work within a global conversation on diabetes burden This suggestion has been effected. The new sentences now read as: 

Global prevalence of diabetes among adults aged 20-79 years in 2021 was estimated by the International Diabetes Federation (IDF) to be 537 million, which is predicted to rise to 783 million by 2045, with the largest increases expected in sub-Saharan African countries [1]. Estimated figures by IDF for Ghana were 329,200 in 2021, representing about 2% of the adult population [1]. However, community-based studies suggest a much higher prevalence than stated [2].

The rising incidence and prevalence of especially type 2 diabetes are due to sedentary lifestyles, obesity, and poor dietary habits [3]. This would bring with it both microvascular (neuropathy, retinopathy and nephropathy) and macrovascular (stroke, myocardial infarction and peripheral vascular disease) complications [1, 3]. The expected increases in diabetes-related morbidity and mortality require corresponding improvements in human and material resources. [page3]

4. Comment: include refs

https://journals.plos.org/plosone/article?id=10.1371/journal.pone.0194677

The suggested reference has been added as reference no. 6 [page3]

 Comment: Even community interventions are also low: https://pubmed.ncbi.nlm.nih.gov/36398761/

The suggested reference has been added as reference no. 7 [page 4]

5. Comment:The rationale is very weak; to pick from one study conducted in the 1990s. 

Please read these: https://journals.sagepub.com/doi/10.1177/13558196221111708#:~:text=Diabetes%20clinics%20for%20outpatients%20are,health%20insurance%20scheme%2C%20the%20NHIS. 

These reference and inferences have been incorporated as part of the rationale under the introduction as references 10 [page 4]

6 Comment: Take note of this reference: https://www.medrxiv.org/content/10.1101/2023.04.19.23288806v1.full

The suggested reference and inference have been added as reference no. 12 [page 5]

7. Comment: Is T2D treatment approach not provided in the standard treatment guidelines 2017.

Whilst it is true that some aspects of the Standard Treatment Guidelines, it is not comprehensive enough. The amended sentence now reads:

“Whilst Standard Treatment Guidelines for Ghana has portions on the management of diabetes, it is not comprehensive and is also restricted for use at the primary and secondary levels. A truly national and comprehensive guidelines for the management of diabetes dedicated to all levels of care was unavailable at the time of this survey.” [page 5]

8. Comment: Add the last 2 paragraphs of the ‘Introduction’

This suggestion has been effected to now read:

“An assessment of the current state of diabetes care was urgently needed to identify the gaps and possible solutions for improvement. A system which is robust in terms of personnel, physical facilities, diabetes care equipment, adequate and affordable medicines and laboratory services, financing and governmental support is needed to address current and future challenges. This study or needs assessment sought to update the human and material resources as well as identify the gaps and unmet needs for comprehensive diabetes care in Ghana since the last update some 25 years ago.” [page 5]

9. Comment: Were the team trained. please include how long and the content of the training 

Yes, the team was trained. The now included statement reads:

“They were trained over 3-day period in community/facility entry protocols, selection of responsible respondents, questionnaire administration, data capture and entry.” [Page 6]

10. Comment: add the questionnaire as a supplementary file 

This has been added as a supporting file

11. Comment:Add a space:

The suggestion on line 202 has been effected in the sentence: “Only 85 (69.7%) out of the 122 surveyed facilities had a dedicated centre…….” [page 8]

12 Comment: in your methods include a section on study variables. 

Write about the kinds of professionals needed for diabetes and provide refs. and indicate that you asked about these in your study 

A subsection on study variables now reads:

“Study variables: Information obtained centered on their current state of diabetes care in terms of facilities and resources as well as existing gaps in managing diabetes. Specific areas assessed included personnel, multidisciplinary teams (trained diabetes doctors/specialists, diabetes educators and nurses, ophthalmic nurses, psychologists, ophthalmologists, foot care specialists/podiatrists dieticians and foot or vascular surgeons) [16], access to medications, access to laboratory services, financing, screening services, management of diabetes complications, and availability/use of diabetes guidelines or protocols as well as support groups.”[pages 6,7]

13. Comment: please introduce all these variables in the methods section 

This has been effected in comment no. 12 above [pages 6,7]

14 Comment: A cross tabulation between these variables and the type of facility will be appropriate 

This suggestion has been effected in tables 2,3 and 4 [Pages 9,10,13]

15. Comment: be consistent with spacing (results) 

This suggested has been complied with throughout the results section 

16. Comment: use upper case

 Please design your tables well. and professionally using excel. 

The tables have been redesigned . They are clearer in the clean document 

17. Comment: why dont the facilities have the required eduipment. the authors should reflect on this question and revise their discussion accordingly 

These possible reasons have been carefully adduced in the discussion section [pages 14-17]

 Comment: The authors repeat most of the results in the discussion, the focus should be on the why 

This section has been incorporated in the discussion section [Pages 14-17]

18 Comment: add the study limitations 

 add the section on the implication of the findings 

2 subsections, study limitations and implications of study findings have now been added. Another sentence has also been added to the conclusion [page 18]

---

## [Decision Letter · Decision Letter 1]

3 Apr 2024

PONE-D-23-31002R1A national audit of facilities, human and material resources for the comprehensive management of diabetes in Ghana-A 2023 updatePLOS ONE

Dear Dr. Yorke,

Thank you for submitting your manuscript to PLOS ONE. After careful consideration, we feel that it has merit but does not fully meet PLOS ONE’s publication criteria as it currently stands. Therefore, we invite you to submit a revised version of the manuscript that addresses the points raised during the review process.

In particular, edit the discussion so that it is not a repetition of the results. Let the section focus on the reasons (why) that explain your findings.

We look forward to receiving your revised manuscript.

Kind regards,

Sandra Boatemaa Kushitor, Ph.D.

Academic Editor

PLOS ONE

Journal Requirements:

**Additional Editor Comments:**

Dear Dr Yorke,

Well done revising this manuscript. Kindly address the minor comments raised by the reviewers and me in the manuscript. Most importantly, strengthen the discussion and delete the repetition of the results in that section. Dedicate the discussion to explaining the results.

Kind regards,

Reviewers' comments:

Reviewer's Responses to Questions

**Comments to the Author**

1. If the authors have adequately addressed your comments raised in a previous round of review and you feel that this manuscript is now acceptable for publication, you may indicate that here to bypass the “Comments to the Author” section, enter your conflict of interest statement in the “Confidential to Editor” section, and submit your "Accept" recommendation.

Reviewer #1: (No Response)

2. Is the manuscript technically sound, and do the data support the conclusions?

Reviewer #1: Yes

3. Has the statistical analysis been performed appropriately and rigorously? 

Reviewer #1: Yes

4. Have the authors made all data underlying the findings in their manuscript fully available?

Reviewer #1: Yes

5. Is the manuscript presented in an intelligible fashion and written in standard English?

Reviewer #1: Yes

6. Review Comments to the Author

Reviewer #1: This study or needs assessment, aimed at updating the human and material resources, identifying the gaps and unmet needs for comprehensive diabetes care in Ghana is well written.

However, there are few questions I have for the authors to answers and comments they have to consider

1. Kindly explained train diabetes doctors, are they diabetologist or medical doctors that have been trained to treat patients with diabetes? How many diabetologist do you have in Ghana?

2. Diabetes educators, what qualifications do they have? (Well trained educators or health care workers trained to educate patients with diabetes)

3. Few typographical errors? Authors should kindly read through the manuscript to correct all grammatical and typographical errors

4. Does the national treatment guideline give comprehensive approach in treating diabetes? What is the target health care workers to use national treatment guideline?

7. PLOS authors have the option to publish the peer review history of their article (what does this mean?). If published, this will include your full peer review and any attached files.

Reviewer #1: No

---

## [Author Response · Author response to Decision Letter 1]

11 Apr 2024

Responses to Reviewers

1. Comment: The discussion repeats a lot of the findings presented in the results. Kindly edit. Present a one paragraph on key findings without stats(if possible) and dedicate the majority of the discussion to explaining the results 

Response: Thanks for the suggestion. This suggestion has been effected. 

References to the detailed results have been edited out or reduced dramatically in the discussion. These have been done as tack changes. (pages 15-18)

2. Comment: Kindly explained train diabetes doctors, are they diabetologist or medical doctors that have been trained to treat patients with diabetes? How many diabetologist do you have in Ghana?

Response: Thanks for the comment. Trained diabetes doctors are doctors who have basic medical qualification and have acquired additional training in diabetes care such as short courses and certificates. They are lumped with diabetologists in this study, who are doctors and have acquired additional specialist qualification in diabetes care or endocrinology. Such diabetologists may be internal medicine physicians, endocrinologist, or family physicians. 

The number of diabetologists including endocrinologists are approximately 20.

Trained diabetes doctors/diabetologists provide specialised diabetes care to patients.

3. Comment: Diabetes educators, what qualifications do they have? (Well trained educators or health care workers trained to educate patients with diabetes)

Response: Thanks for the comment. 

The term ‘diabetes educators’ as used in this study loosely refers to health care workers who are certified diabetes educators or have received some form of training to provide diabetes education (without certification).

This was done considering the limited number of certified diabetes educators in Ghana. 

4. Comment: Few typographical errors? Authors should kindly read through the manuscript to correct all grammatical and typographical errors

Response: Thanks for the comment.

Grammatical and typographical errors have been corrected throughout the manuscript and highlighted a track changes. 

5. Comment: Does the national treatment guideline give comprehensive approach in treating diabetes? What is the target health care workers to use national treatment guideline?

Response: Thanks for the comment. 

The Standard Treatment Guidelines (STGs) does not give comprehensive approach in treating diabetes. It covers treatment approaches for a wide range of diseases.

Although, the ministry of health suggests that the STGs could be used at all levels of care, in practice, it is used mostly at the primary and secondary levels of care (health centres, district, and regional hospitals). This has been highlighted in the discussion section on page 17. Tertiary level practitioners will mostly rely on other guidelines from bodies such as the American Diabetes Association. (page 17-18)

6. Comment : Can you add one possible recommendation at facility level (Abstract conculsion) 

Response: Thanks for the comment.

The new conclusion of the Abstract reads: “An urgent multisectoral collaboration, including prioritisation of resources at the facility level, to promote and achieve acceptable comprehensive diabetes care is required.” Page 3

7. Comment: Please indicate that diabetes is structured to start at the health centers. CHPS have not been designed to treat NCDs according to their protocols 

Response:Thanks for the comment.

The suggestion has been incorporated. The new sentence reads: “Notwithstanding that CHPS compounds are not designated to treat diabetes, many of the other facilities do not have specialised centres for diabetes care. Most diabetes patients are therefore treated as part of the general clinic and hospital care.” Page 4

---

## [Editor Report · Decision Letter 2]

30 Apr 2024

A national audit of facilities, human and material resources for the comprehensive management of diabetes in Ghana-A 2023 update

PONE-D-23-31002R2

Dear Dr. Yorke,

We’re pleased to inform you that your manuscript has been judged scientifically suitable for publication and will be formally accepted for publication once it meets all outstanding technical requirements.

Kind regards,

Sandra Boatemaa Kushitor, Ph.D.

Academic Editor

PLOS ONE
---

## [Editor Report · Acceptance letter]

7 May 2024

PONE-D-23-31002R2 

PLOS ONE

Dear Dr. Yorke, 

I'm pleased to inform you that your manuscript has been deemed suitable for publication in PLOS ONE. Congratulations! Your manuscript is now being handed over to our production team.

Kind regards, 

on behalf of

Dr. Sandra Boatemaa Kushitor 

Academic Editor

PLOS ONE